# Ecological Evaluation of the Sustainability of City Forests

**Milijana Cvejić [1,*], Marko Joksimović [2], Jelena Tomićević-Dubljević [3], Ljubinko Rakonjac [1], Milan Medarević [3] and Vladimir Malinić [2]**

1 Institute of Forestry, Kneza Višeslava 3, 11000 Belgrade, Serbia
2 Faculty of Geography, University of Belgrade, Studentski trg 3/3, 11000 Belgrade, Serbia
3 Faculty of Forestry, University of Belgrade, Kneza Višeslava 1, 11000 Belgrade, Serbia
* Correspondence: minacvejic01@gmail.com

**Abstract:** The Košutnjak forest in the city of Belgrade, Serbia, with an area of 259 ha, provides ecological and social benefits to its inhabitants, but its composition has changed in the last 20 years: forest areas have decreased, people have become irresponsible towards the forest and forest soil, and forest degradation is evident. The question is whether the forest has the potential to regenerate and survive. The horizontal assessment of attributes was carried out using data from the official forest database of the administrative unit "Košutnjak (2007–2016)", which, in conjunction with the basic forest, defines indicators of change, stability, and self-renewal, which assume sustainability and can be a useful tool for sustainable forest management. The attributes and indicators are processed on a three-level alphanumeric scale in Microsoft Excel, and the data collected and analyzed are mapped using ArcGis 9.3. The ability of forests to survive without human intervention was evaluated using the EEFS method of ecological assessment of forest sustainability, which was used for the first time in this study. The results showed that forest change was significant, stability was medium, and self-renewal was low on most sections, so forest sustainability was rated as unlikely. The EEFS method used provided results that can form the basis for a forest management strategy in the city and a platform for the long-term monitoring of forest condition.

**Keywords:** attributes; Belgrade; ecological assessment; Košutnjak forest; indicators; sustainability





## 1. Introduction

Forests located in or near cities (city forests) are important because of exposure to intensive degradation caused by human activities and climate change. This decreases the vitality of forest ecosystems and changes the basic (most stable) forest type (determined by edifying species and soil). These changes are assumed to lead to a decrease in the stability of forest ecosystems and, thus, endanger their self-regeneration. Forests are resilient ecosystems; however, their capacity to resist environmental changes is limited, and sometimes when the limits are crossed, they cannot recover. Therefore, sustainable forest management implies that the capacity of forests and their role to otherwise absorb environmental changes have to be monitored, while human activities have to be directed towards ensuring the maximum level of stability.

Forest in the cities are evidently decaying and drying up, without which the survival of the cities is impossible. The research examined the sustainability of the forest, which can be seen through its changes over time, preserved stability, and ability to self-regenerate, without human intervention. Urban and peri-urban forests provide numerous benefits for society. These include moderating the climate; reducing energy use in buildings; sequestering atmospheric carbon dioxide; improving air and water quality; mitigating rainfall run-off and flooding; providing an aesthetic environment and recreational opportunities; enhancing human health and social well-being; and lowering noise impacts [1–4]. To sustain or enhance the benefits of urban and peri-urban forests for society, it is important to analyze forest attributes and changes and to understand how the existing forest attributes

affect sustainable urban forest management. In addition, urban and peri-urban forests are usually part of a protected area management regime whose use is regulated by different legislation and regulative frameworks [5–8]. A legislative framework is defined by different laws, laws and regulations on a national or local level [5,7], while an institutional framework represents all institutional and organizational settings [9,10]. Management of these areas can be given both to the public and private sector and can be divided on the basis of who makes decisions and can be held responsible [11–13]. Forests are resilient ecosystems; however, their capacity to resist environmental changes is limited, and sometimes when the limits are exceeded, they cannot recover; therefore, sustainable forest management implies that the capacity of forests to absorb environmental changes has to be monitored, and human activities have to be directed towards ensuring the maximum level of stability.

The research aimed to assess the impact of changes in the plant composition of stands on their stability and self-regeneration capacity in the of Košutnjak forest, city forest complex (259 ha) in Belgrade, capital of Serbia. Furthermore, the general goal of the research was to define indicators and apply them to Košutnjak in order to gain a deeper insight into the concept of sustainable forest management. In order to observe different impacts and changes relative to the potential of this forest to be managed sustainably, we developed a new method—Ecological Evaluation of Forest Sustainability (EEFS). Changes in the stand composition were analyzed based on data obtained from the forest management plan. They were used to compare the primary tree species with recent (current) vegetation and assess the stability of current stands. Based on the obtained indicators, we could predict the actual self-regeneration capacity of the study stands of the urban forest.

Regarding main hypothesis, the baseline was the assumption that the basic ecological forest type is potentially the most stable form of forest in a given area and the stable forests contribute to the stability of the entire ecosystem as well as prevent soil erosion and loss of nutrients, regulate the level and quantity of water in the soil, reduce wind force, maintain a favorable atmospheric composition, and make the environment more resilient. We also assumed that the primary type of forest vegetation had changed to such an extent that the stability of forest ecosystems and their self-regeneration, as goals and prerequisites of biodiversity, were endangered.

## 2. Literature Review

According to the Helsinki Resolution H1 [14], adopted at the Second Ministerial Conference, sustainable management means "the management and use of forests and forest lands in a manner and at a rate that maintains their biological diversity, productivity, regenerative capacity, vitality, and their potential to provide relevant ecological, economic, and social functions at local, national, and global levels, now and in the future, and does not cause damage to other ecosystems" (MPCFE, Helsinki 1993) [15]. The Second Ministerial Conference on the Protection of Forests in Europe, held in Helsinki in 1993, provided the basis for a pan-European approach to global goals such as sustainable forest management, conservation and appropriate enhancement of forest biodiversity, and long-term adaptation of European forests to climate change, through the Universal Declaration and four resolutions. The logical continuation was the definition of the pan-European criteria and indicators for sustainable forest management at the Third Ministerial Conference in Lisbon in 1998 [16]. All international initiatives launched to assess progress towards sustainable forest management define principles, criteria, and indicators. Principles are explicit elements of a goal, such as sustainable forest management. Criteria are defined as aspects of forest management that are considered important and against which the success of sustainable forest management can be assessed. They are used to characterize or define the essential elements, conditions, or processes by which sustainable forest management can be judged. Indicators are quantitative, qualitative, or descriptive characteristics that, when measured and monitored regularly, indicate the direction of change. More specifically, criteria define the main elements of sustainable forest management. Indicators, on the other hand, provide a basis for assessing the current status of forests. When international

criteria and indicators (C&I) are combined with specific national targets, they provide the basis for assessing progress toward sustainable forest management. The development and implementation of C&I allows, on the one hand, a deeper insight into the concept of sustainable forest management and, on the other hand, its "transfer" to an operational level in terms of application in forest management.

National classification and labelling systems are not complete until forest management classification and labelling systems are defined and implemented at the forest area level. P&Is at the forest area level are determined not only by social and economic considerations, but also by other factors such as forest type and topography. Therefore, P&Is of forest management at the forest area level may vary from forest area to forest area in a country. They may also differ over time, depending on prevailing conditions or management priorities and objectives in the area. Assessing progress in sustainable forest management over time clearly demonstrates the importance of developing and implementing national criteria and indicators. This progress is evaluated using measurable indicators and in relation to the goal set. The criteria and indicators are a useful tool for sustainable forest management because they provide relevant information needed to develop national forest policies and assessments [17]. National criteria and indicators are only complete if they are defined and implemented at a lower level, i.e., at the forest area (forest ownership) and forest management (community) level [18].

Few authors have studied the ability of forest areas to provide recreation services [19–21]. However, these studies never went beyond the local planning level or rarely reached the regional or state level. The analyses of forest areas (in these research studies) were mostly related to their use and consequently to their change in the form of area change, i.e., fragmentation or reduction. In addition to the use of the forest and forest land to satisfy human needs, important research studies deal with the construction of various facilities and buildings such as residential, accommodation, catering, or historical buildings inside an urban forest and on its edge, as shown by the planning in the papers [22–24]. The experience of forest professionals with similar problems was used for the applied method [25–29]. Planning, evaluation, and mapping were the activities that helped to select, evaluate, and provide results from the collected data, based on which further decisions were made for urban forests in Belgrade [27–32].

## 3. Method

### 3.1. Study Area

Forest management in Serbia is based on data on the condition of the forest stand and habitats obtained from regular forest inventory monitoring. Due to its location and spatial planning (Master Plan Belgrade 2021 (MP)) [33], the forest of the administrative unit "Košutnjak" (MU) belongs to the urban forests of the city of Belgrade. The forest wedge-shaped hill extends to the city center. The value of this forest lies in its rich biological diversity, well-developed landscape, and favorable location. Košutnjak forest was declared a Strict Nature Reserve and, together with the pedunculate oak and hornbeam forest within it, was declared a Protected Primeval Forest by the Institute for Nature Conservation of Serbia (INCS) (2008) (Figure 1). According to archival records, Košutnjak was a popular picnic spot (Figure 2) for citizens after the Second World War [34]. Even today, this place is visited by many people, which over time has resulted in the forest edge, both on the outer edge and inside the forest around the clearings, being the weakest zone of the forest, which is under constant pressure of slow but relentless urbanization. The new state of the forest from 1999 to the present is satisfactory; urbanization has begun to bring order to the forest so that both forest vegetation and human activities in the forest do not interfere with each other.

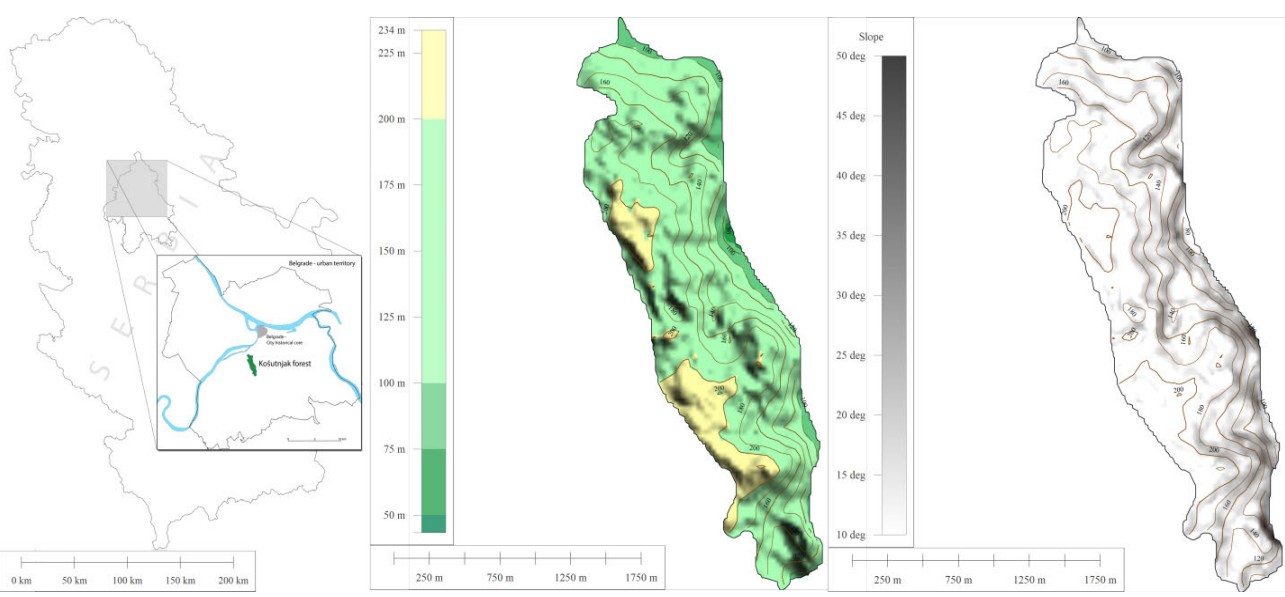

**Figure 1.** Location, topography and slope angle of the terrain of Košutnjak Forest in Belgrade.

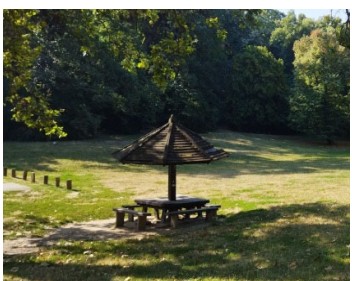

**Figure 2.** Picnic area in Košutnjak forest.

Due to its exceptional natural and cultural–historical values, the forest "Košutnjak" was declared a cultural heritage of exceptional importance for Serbia by the Institute for the Protection of Cultural Monuments of Serbia (IPCMS). Due to its favorable geographical position, climatic and geomorphological characteristics, and richness of flora and fauna (INCS), it was declared a natural monument. Based on the criteria, condition, and potential of the forests and wooded areas in MU "Košutnjak", the following priority functions have been defined: first degree soil protection; first degree tourist recreation centre (Figure 3); strict nature reserve. The primaeval forest of English oak and hornbeam near the "Hajduk drinking fountain" (Figure 4) in Košutnjak was declared a strict nature reserve. On the satellite image Kosutnjak can be seen as an isolated island of vegetation within the built-up urban area (Figure 5).

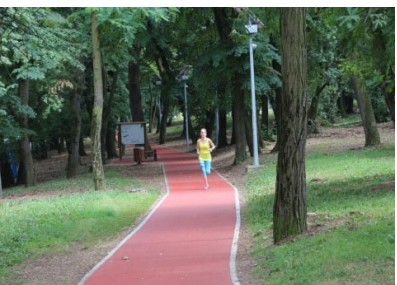

**Figure 3.** Recreation in Košutnjak forest.

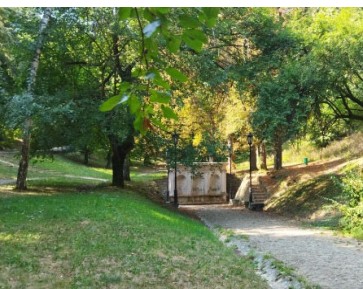

**Figure 4.** Primeval forest near "Hajduk's drinking fountain" in Košutnjak forest.

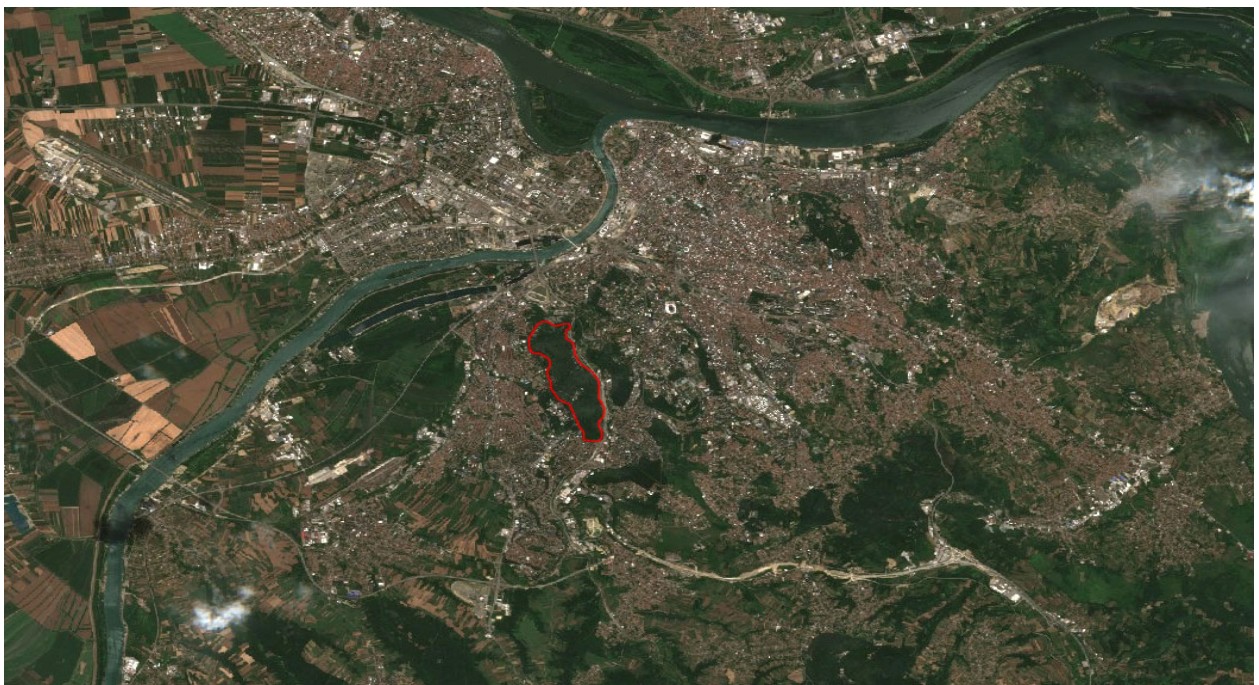

**Figure 5.** Satellite imagery of forest Košutnjak in Belgrade, 29 June 2022.

The landscape of MU "Košutnjak", with an area of 259 ha and diverse vegetation, is very diverse. It consists of steep slopes intersected by well-formed valleys. The slopes and depressions are steep and exceed 70%. The range between the lowest and the highest point is 75–216 m above sea level, which makes this management unit suitable for recreational purposes and nature excursions.

The change in the composition of the forest stand in MU "Košutnjak", City of Belgrade, became evident and threatened the survival of the studied forest ecosystem. Monitoring of the state of vegetation in the Košutnjak forest (forest management plan for MU "Košutnjak") over a period of 10 years, for which an official tree species register was available (2006 to 2017), revealed changes and deterioration in the composition of stands, their lower vitality and deteriorated health of forest ecosystems. This means that forests are not as stable as they once were. They are in a weaker state of health and can no longer regenerate on their own. Therefore, it is necessary to provide them with appropriate maintenance measures.

The forest of MU "Košutnjak" belongs to the Sava–Danube forest area. The forest area consists of the following different forest communities: pedunculate oak and ash forest (Fraxino-Quercetum roboris) on moist semi-gley and dry gley soils; pedunculate oak, Hornbeam, and ash forest (Carpino-Fraxino-Quercetum roboris) on semi-gley soils; the pedunculate oak, hornbeam, and sessile oak forest (Carpino-Quercetum robori-cerris) on semi-hill soils; eutric cambisol and brown lessivé soils, the typical Hungarian oak and Turkey oak forest (Quercetum frainetto–cerris typicum) on brown Lessivé soils; the

sessile oak and Turkey oak forest (Quercetum petraeae—cerris) on loess, silicate rocks, and limestone; the sessile oak–hornbeam forest (Querco-carpinetum moesiacum) on brown and leached brown soils; and the sessile oak–hornbeam forest (Carpino-Quercetum petraeae–cerris) on soils over loess and acid silicate rock (Figure 6).

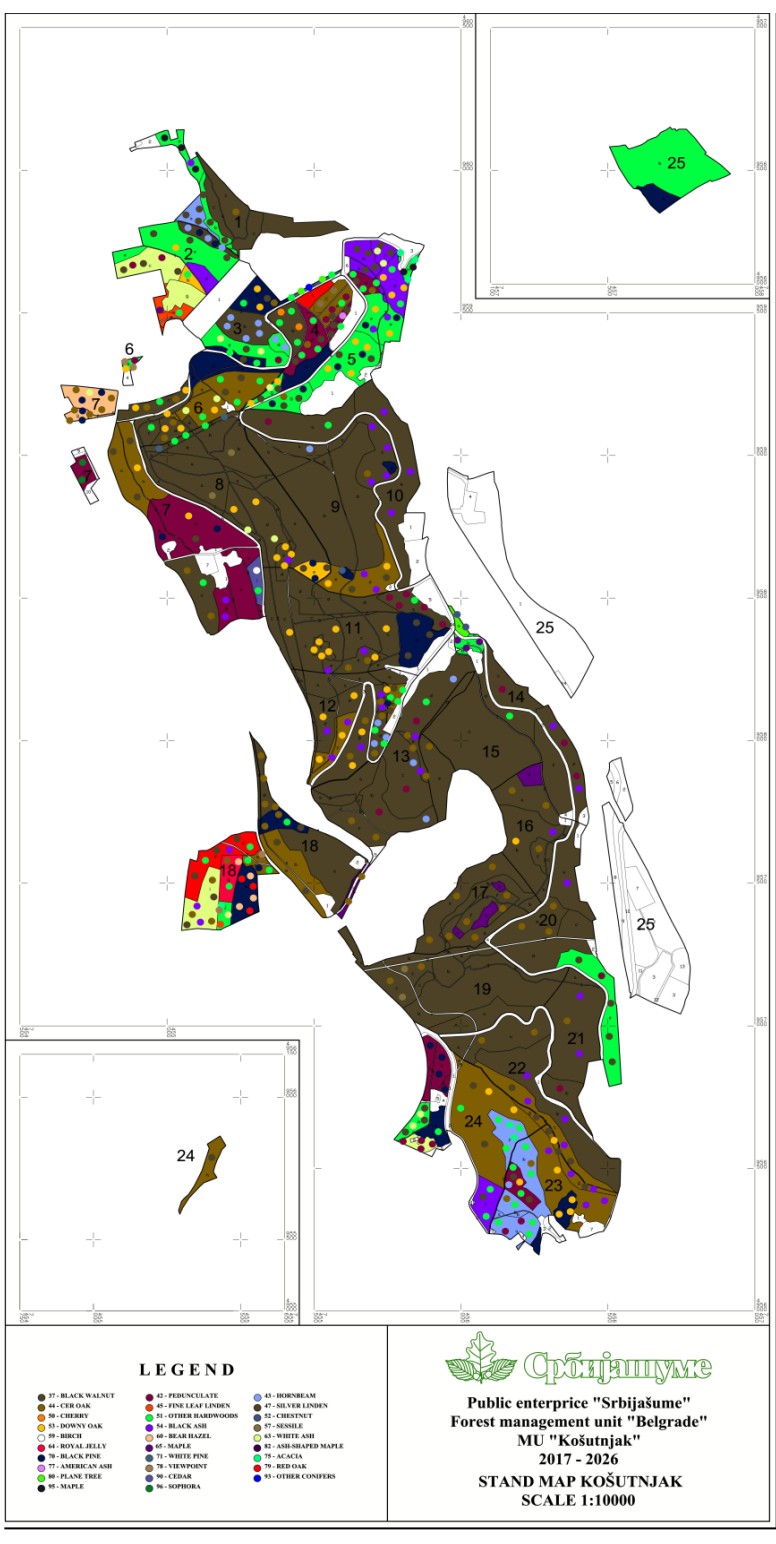

**Figure 6.** Stand map of Košutnjak forest. Numbers—Forest department.

An analysis of forestry books that monitor and revise forest vegetation every 10 years shows that primary vegetation has changed to a greater or lesser extent due to social changes (wars, fires), entomological, and phytopathological damage, and so on. These changes can be evidenced by historical sources [34], relevant documents, and forest management plans for this management unit.

### 3.2. Method and Data Collection

The sustainable management assessment method was applied to the study of the 259 ha Košutnjak forest complex, and various impacts and changes were identified in relation to the potential of this forest to be sustainably managed. The aim of the research was to evaluate the impact of changes in the plant composition of the stands on their stability and self-regeneration capacity in the Košutnjak forest.

As part of the research, the Ecological Evaluation of Forest Sustainability (EEFS) method was developed. The horizontal assessment of forest stands over a period of 10 years for the management unit "Košutnjak", 2006–2017, public enterprise (PE) "Srbijašume", was carried out using a manual method with the introduction of the indicators of change, stability, and self-regenerative capacity. Microsoft Excel was used to process the collected data. The results were presented with ArcGis 9.3, using the software for mapping the collected and analyzed data. The Global Mapper v18.2 application was used to create maps using Aster DEM free maps. The tools used included the create contours, slope direction shader, and gradient shader.

The indicators applied in this research can be useful tools for sustainable forest management and could be used to obtain relevant data and information for the development of a national forest policy and to assess the sustainability of urban forests. Sustainability is a natural characteristic of forests. It depends on stability, the ability to self-regenerate, and the degree of change. Moreover, the sustainability of the whole city depends on the sustainability of urban forests [19]. Evaluation of indicators and analysis of risk factors (diseases, pests, human impact, wind, floods, fires, etc.) show the degree of change and allow to evaluate the stability and self-regeneration capacity of the Košutnjak urban forest (Figure 7).

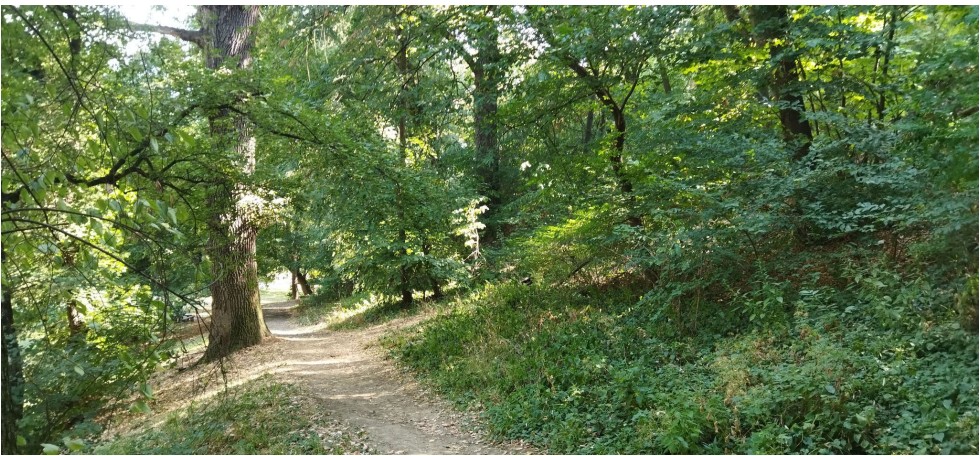

**Figure 7.** Some pathways in Košutnjak forest appear as human impact.

The primary data were obtained from the forest management plan of MU "Košutnjak" for the period 2006–2017, PE "Srbijašume". It contains data on the main environmental features (Figure 8).

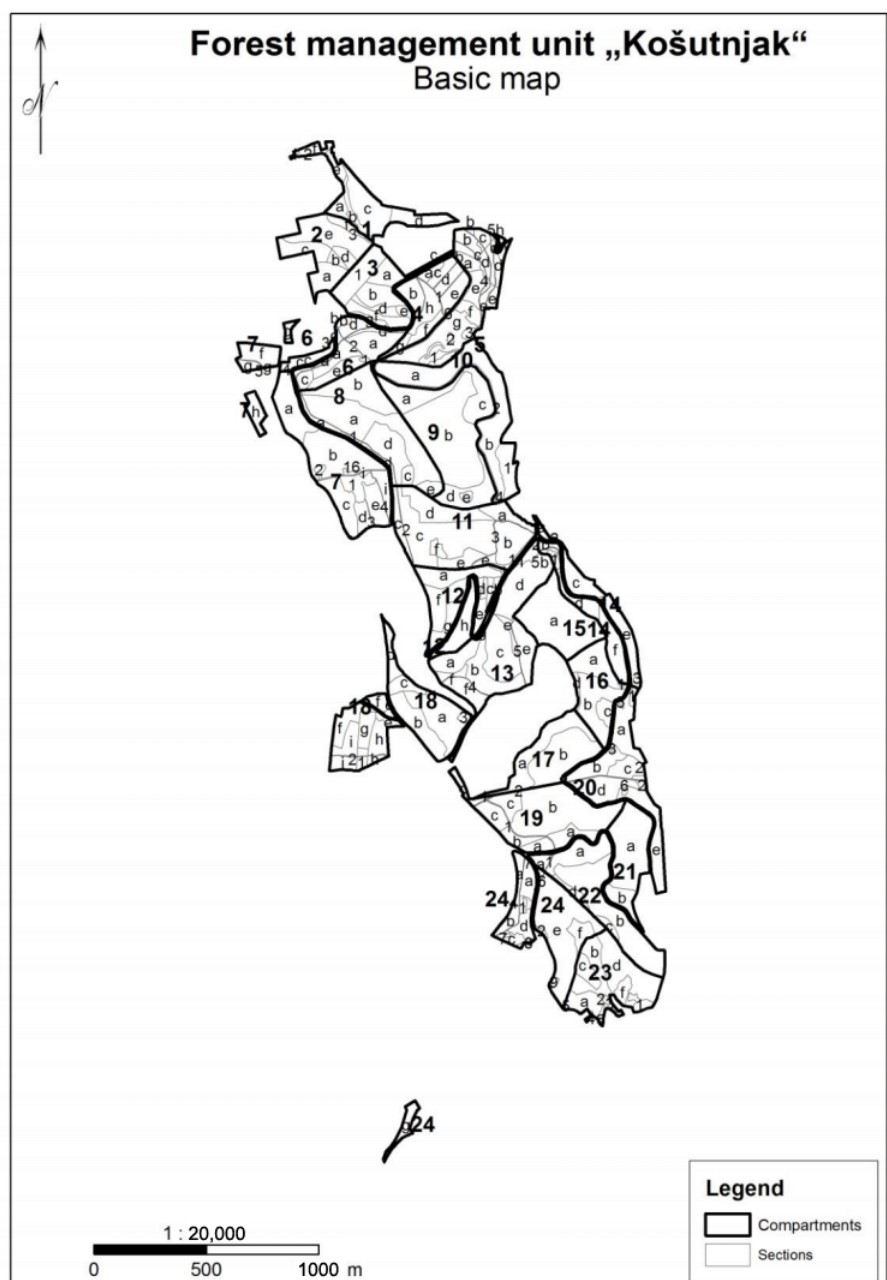

**Figure 8.** Forest management map. Numbers—Forest department, Letters—Department subdivision.

The data collection was conducted in two phases. In the first phase of data collection on vegetation and habitat for the study of the organization of the forest management system, it was defined on the basis of the forest register and the data of the public company (PE) "Srbijašume". First, the change in stand composition was studied at the section level by comparing the basic forest type with the corresponding current vegetation, taking into account the stand density (%) and the tree species and their mixtures (%). The relationship between the basic forest type and the current forest was then correlated and scored on a three-point scale as slight, moderate, or substantial change, as shown in Table 1.

**Table 1.** Assessment of the change of stands in MU "Košutnjak", 2006–2017, PE "Srbijašume".

| Compartment | Section | Attributes of Change | | | | | | "Indicator of Change" |
|---|---|---|---|---|---|---|---|---|
| | | Basic Forest | | Recent Forest | | | | |
| | | Basic Forest Vegetation | Recent Forest Vegetation | Forest Density | | Tree Species | Mixture | Significant 3 Medium 2 Slight 1 |
| | | | | (%) | | | (%) | |
| | | | | Assessment | | | Assessment | |
| **1** | **2** | **3** | **4** | **5** | | **6** | **7** | **8** |
| 1 | a | Forest of sessile oak and hornbeam | Forest of hornbeam-field maple-linden | 0.5–0.6 | 3 | Hornbeam field maple linden | 0.7   1 | 3/(4–7) Column 3 relative to columns 4 to 7 |
| | | | | 0.7 | 2 | | 0.2   2 | |
| | | | | 0.8–0.9 | 1 | | 0.1   3 | |
| Assessment 1a | | | | 2 | | | 1 | Medium 2 |
| 1 | b | Forest of sessile oak, hornbeam and Turkey oak | Forest of linden, o.h.s., field maple | 0.5–0.6 | 3 | Linden, o.h.s., field maple | 0.4   2 | 3/(4–7) Column 3 relative to columns 4 to 7 |
| | | | | 0.7 | 2 | | 0.4   2 | |
| | | | | 0.8–0.9 | 1 | | 0.1   3 | |
| Assessment 1b | | | | 2 | | | 3 | Significant 3 |

The assessment was conducted in three steps. The first step was to obtain the forest type for each division and section of the forestry analyzed.

In the second step, the stability of the stands and their ability to self-regenerate were assessed. The following attributes were used to evaluate the stability of the stands: age, stand density or canopy size, productivity, weed stocking, health status, degree of vulnerability depending on the type of harmful impact, and risk factors. In the third step, the ability of the stands to self-regenerate was evaluated. The Košutnjak complex is overall (or mostly) characterized by over-maturity of autochthonous tree species and suppression of species that should carry the natural recovery potential—primarily various oak species (pedunculate oak, sessile oak, Hungarian oak, Turkey oak, downy oak, and large-leaved downy oak). The assessment of self-regeneration capacity showed primarily that there was sufficient compatibility between the habitat and the recent vegetation, i.e., the up-graders as defined in previous typological studies in the Košutnjak forest complex [35]. Other important indicators were age (the developmental stages suitable for analysis are maturation and later physiological maturity for fruiting, or at the other end, death), weediness at a given age, and origin. Based on the estimated stability and the previously evaluated change in stand composition, it was possible to predict the self-regeneration capacity of the studied stands, as shown in Tables 2 and 3.

In production and commercial forests, the life span of English oak is about 200 years and that of sessile oak is 120–140 years. Very rare English oak trees in Serbia are older than 150 years for many reasons. In particular, urban forests in Belgrade survived the destruction during WW1 and WW2 and the bombing of Belgrade in 1999. With the material crisis in the 1990s of the 20th century and until today, the trees still had the potential to survive on their own without the help of man. It was expected that the result of the forest change analysis would show whether the forest has changed substantially from its original composition to today in terms of its habitat potential—Zeno-ecological affiliation, because it is assumed that the original forest with suitable forest vegetation is the most stable and sustainable forest. Due to the mentioned and other reasons in the area, the average age of a mature forest stand in Košutnjak is about 80 years, if it is of natural origin, and in 1954, reforestation began along the edges of the planted forests, mostly on agricultural land expropriated from the owner.

**Table 2.** Assessment of the self-regeneration capacity of the stands 1a and 1b in the MU "Košutnjak", 2006–2017, PE "Srbijašume".

| Attributes | Range | | Condition | Assessment | |
|---|---|---|---|---|---|
| Age (year) | 1–20 20–40 >40 | Young tree Mature tree Adult tree | 64 | Adult tree | 1 |
| Stand canopy | 0.5–0.6 0.7 0.8–0.9 | Thin Dense Very dense | 0.7 | Dense canopy | 2 |
| Productivity (m³/ha) | >30 21–30 5–20 | High Medium Low | 10.2/1.5 (6.8) | Low productivity | 3 |
| Weediness | - | Low Medium High | - | Lowweediness | 1 |
| Health condition | | Good Moderate Poor | - | - | - |
| Vulnerability | | Low Medium High | - | - | - |
| Adverse effect | | Nothing Man Phyto pathologic damage Fire | - | - | - |

**Table 3.** Indicators of change, stability and self-regeneration capacity in the MU "Košutnjak", 2006–2017, PE "Srbijašume".

| Section 1a | | | | | | | | | | | |
|---|---|---|---|---|---|---|---|---|---|---|---|
| Indicator of Change | | | | Indicator of Stability | | | | Indicator of Forest Self-Regeneration | | | |
| At | As | At | As | At | As | At | As | At | As | At | As |
| Significant | 3 | | | High | 1 | | | High | 1 | | |
| Medium | 2 | Medium | 2 | Medium | 2 | Medium | 2 | Medium | 2 | Medium | 2 |
| No change | 1 | | | Low | 3 | | | Low | 3 | | |
| The sustainability of Section 1a is unlikely 3. | | | | | | | | | | | |
| Section 1b | | | | | | | | | | | |
| Indicator of change | | | | Indicator of stability | | | | Indicator of forest self-renewablity | | | |
| At | As | At | As | At | As | At | As | At | As | At | As |
| Significant | 3 | Significant | 3 | High | 1 | | | High | 1 | | |
| Medium | 2 | | | Medium | 2 | | | Medium | 2 | | |
| No change | 1 | | | Low | 3 | Low | 3 | Low | 3 | Low | 3 |
| The sustainability of Section 1b is unlikely 3. | | | | | | | | | | | |

At—attributes. As—assessment.

In the third step (Tables 2 and 3), we analyzed the current state of vegetation, i.e., recent forest, which is composed of the following (Table 1, columns 4–7): recent forest vegetation, tree density, tree species, and mixture. In Section 1a, the recent vegetation (Table 1, column 4) consists of a tall forest with hornbeam, field maple, and linden, which in its overgrowth mixture (column 7) represents 0.7%, while the three most represented tree species are the mixture with hornbeam 0.7%, field maple 0.2%, and linden 0.1%.

The main limitation of the methodology used is the collection and analysis of the data. This was a sensitive and lengthy process. Universal application of the method is

possible because the method offers the possibility to analyze the smallest ecologically stable fragments of forest vegetation at three levels: large, medium, and small, through three states: modified, stable, and renewable; it is possible to quickly and easily evaluate the future resilience of the entire forest in the city. The method can also be applied to other small forests remaining within the city limits.

## 4. Results

The research results show that the stand composition of the Košutnjak forest has changed significantly on 63.09% of its forest area. Its stability has also decreased. A high level of forest stability was found only on 1.71% of the area. These results raise doubts that the forest can rejuvenate itself. According to the results, the studied area of the Košutnjak forest had a low self-regeneration capacity (64.55%) on more than 60% of the area. The changes were more pronounced on the forest edges near the urban areas. On the other hand, a medium stability was present in all parts of the forest, both within and at the edge.

Our research results indicate that the changes in plant composition have a negative impact on stand stability, thus reducing the self-regeneration capacity of the studied forest complex. The research also confirmed the original assumption that the stability of the forest complex "Košutnjak" was reduced due to the significant changes in the stands (observed on 56.81% of the section area) and was classified as medium on 66.66% of the section area (Table 3), while the self-regeneration capacity of most sections (59.84%) was low—insufficient in terms of sustainability.

First, the forest change assessment was conducted. Comparing the vegetation in Section 1a, which belongs to the basic forest type (column 3), with the current forest vegetation (column 4), it was found that the sessile oak has completely disappeared, leaving only the hornbeam, which now dominates in the mixture with other species with 0.7%; so, we decided that the forest change is classified as medium, with a value of 2 on a scale of 1 to 3 (Table 1). In Section 1b, the change is significant because it is a basic forest of sessile oak, hornbeam, and English oak, and the species present in the recent forest vegetation of this section are linden, hornbeam, and field maple. There are no more forest species that were present in the basic forest; they have been replaced by other forest species, so we concluded that the change in forest stands in Division 1b is significant and that the rating of change is 3 on a scale of 1 to 3. Based on the data of the assessment of change, sustainability, and self-renewal of Division 1a, very low productivity (6.8 m$^3$/ha), dense canopy (0.7), mature forest (64 years), and weak weediness, it is estimated that the expected self-renewal is impossible on the analyzed forest section with the worst score of 3, because the productivity of the forest is very low (Table 2) and it cannot be expected to return to its original state.

The next step was to evaluate the stability of the current forest in Table 3:

1.  Age of the forest (1–20, 20–40, >40) is 64 years; value of 1;
2.  Stand canopy cover (thin 0.5–0.6; dense 0.7; very dense, 0.8–0.9); value of 1;
3.  Productivity is 10.2 m$^3$/1.5 ha, which corresponds to 31.73 m$^3$/ha (>30, 21–30, 5–20) (>30); productivity value of 1;
4.  Weediness (low, medium, high) is medium; value 2;
5.  Health status (good, moderate, poor) is without data;
6.  Susceptibility (low, medium, high) is without data;
7.  Adverse effects (no, by humans, by phytopathological damage, fires) is without data.

Based on the data on the assessment of changes, sustainability and self-renewal of Division 1b, it is estimated that despite the estimated good productivity (31.73 m$^3$/ha), very dense canopy (0.8–09), mature forest (64 years), and medium weediness, the expected self-renewal in the analyzed forest division is impossible, with the worst score of 3, because the change in the forest is significant (Table 1) and it cannot be expected to return to its original state. The baseline elements (levels) within which the change in the basic type of the analyzed Košutnjak forest was estimated were compartment and section, forest type, and recent vegetation (Table 1). The stability of the forest was estimated and ranked on

a three-level scale (high, medium, and low) by analyzing the following attributes: age (years), stand canopy (thin 0.5–0.6, dense 0.7, very dense 0.8–0.9), productivity ($m^3$/ha), weediness (low, medium, and high), health (good, moderate, poor), vulnerability (low, medium, and high), adverse effects (human impact, phytopathological damage, and fire) (Tables 2 and 3). The self-regeneration capacity of the forest ecosystem was evaluated by correlating change and stability of the forest ecosystem. It was then ranked on a three-level scale (high, medium, and low) (Table 4).

**Table 4.** Table of change, stability, and self-regeneration capacity in the MU "Košutnjak" (forest management plan of the MU "Košutnjak", 2006–2017, PE "Srbijašume".

| Indicator of Change | Quantity Number of Sections Section Area (%) Section Area (ha) | Indicator of Sustainability | Quantity Number of Sections Section Area (%) Section Area (ha)/ | Indicator of Self-Regeneration Capacity | Quantity Number of Sections Section Area (%) Section Area (ha) |
|---|---|---|---|---|---|
| Significant | 76 63.09% (163.42) | Low | 41 21.57 (55.88) | Low | 80 64.55 (167.20) |
| Moderate | 56 36.91% (95.62) | Medium | 88 76.72 (198.74) | Medium | 46 31.89 (82.60) |
| No change | 0 0 (0) | High | 31.71 (4.42) | High | 63.57 (9.24) |
| Total | 132 sections 100 (259.04) | Total | 132 100 (259.04) | Total | 132 100 (259.04) |

Table 4 shows the results of the evaluation of the three indicators (change, stability, and self-regenerative capacity) with the estimated values (significant, moderate, and no change) or (low, medium, and high) followed by the quantities (for each indicator, we have indicated the number of affected sections, the area of all sections per estimate (%) and the area of all sections per estimate (ha)).

Figure 9 shows the change in the basic forest type. It can be seen that the forest is significantly altered (red) on 63.09% of the area, mainly in the central parts of the forest, while it is moderately altered (yellow) on 36.91%, mainly in the southern parts of the forest. There are no parts that are not affected by changes (green): 0%.

Figure 10 shows the stability of the forest vegetation. Most of the forest (marked in yellow), 76.72%, has medium stability. Low stability (marked in red) was observed in 21.57% of the forest area, in fragments along the central part of the forest, towards NS. The northern edge of the forest shows high stability only on 1.71% of the area (marked in green).

Figure 11 shows the self-regeneration capacity of forest vegetation. Most of the Košutnjak forest, or 64.55%, has a low self-regeneration capacity (marked in red). A mean self-regenerative capacity (marked in yellow) was observed at 31.89%, mainly at the southern edge of the forest. The high self-regenerative capacity of the Košutnjak forest was found on 3.57% of the area in the northern parts of the forest (Table 5).

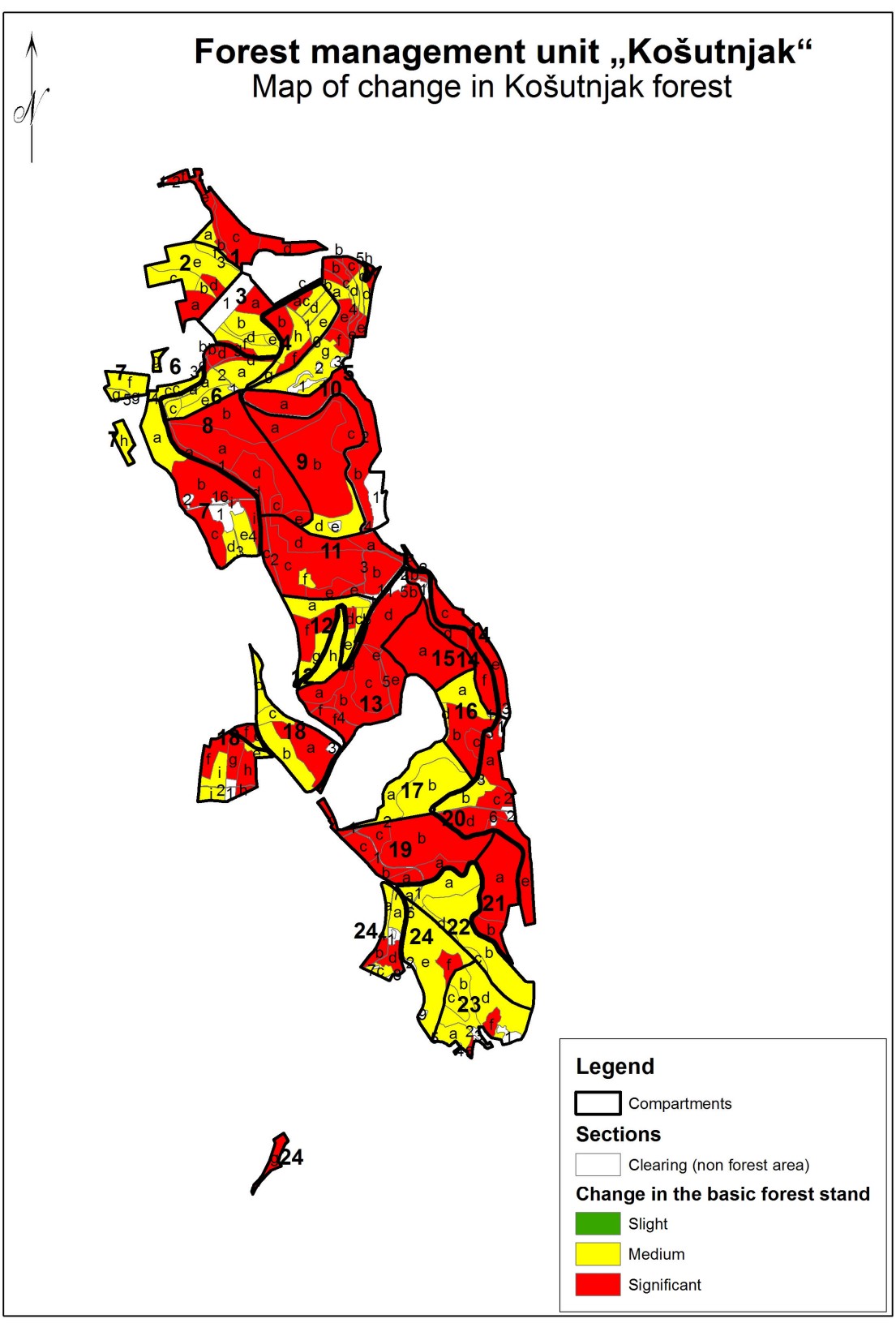

**Figure 9.** Map of changes in the Košutnjak forest. Numbers—Forest department, Letters—Department subdivision.

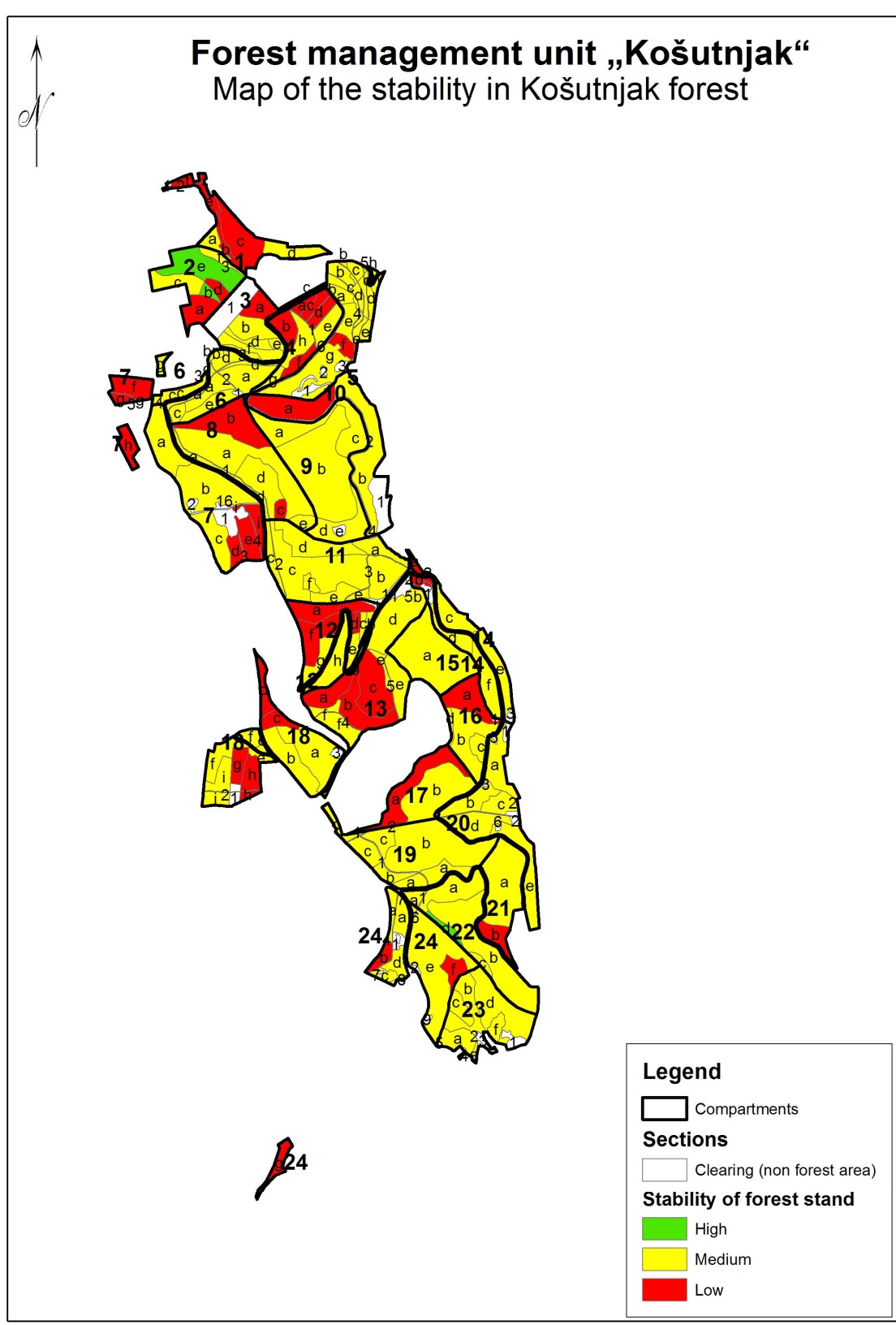

**Figure 10.** Map of stability in the Košutnjak forest. Numbers—Forest department, Letters —Department subdivision.

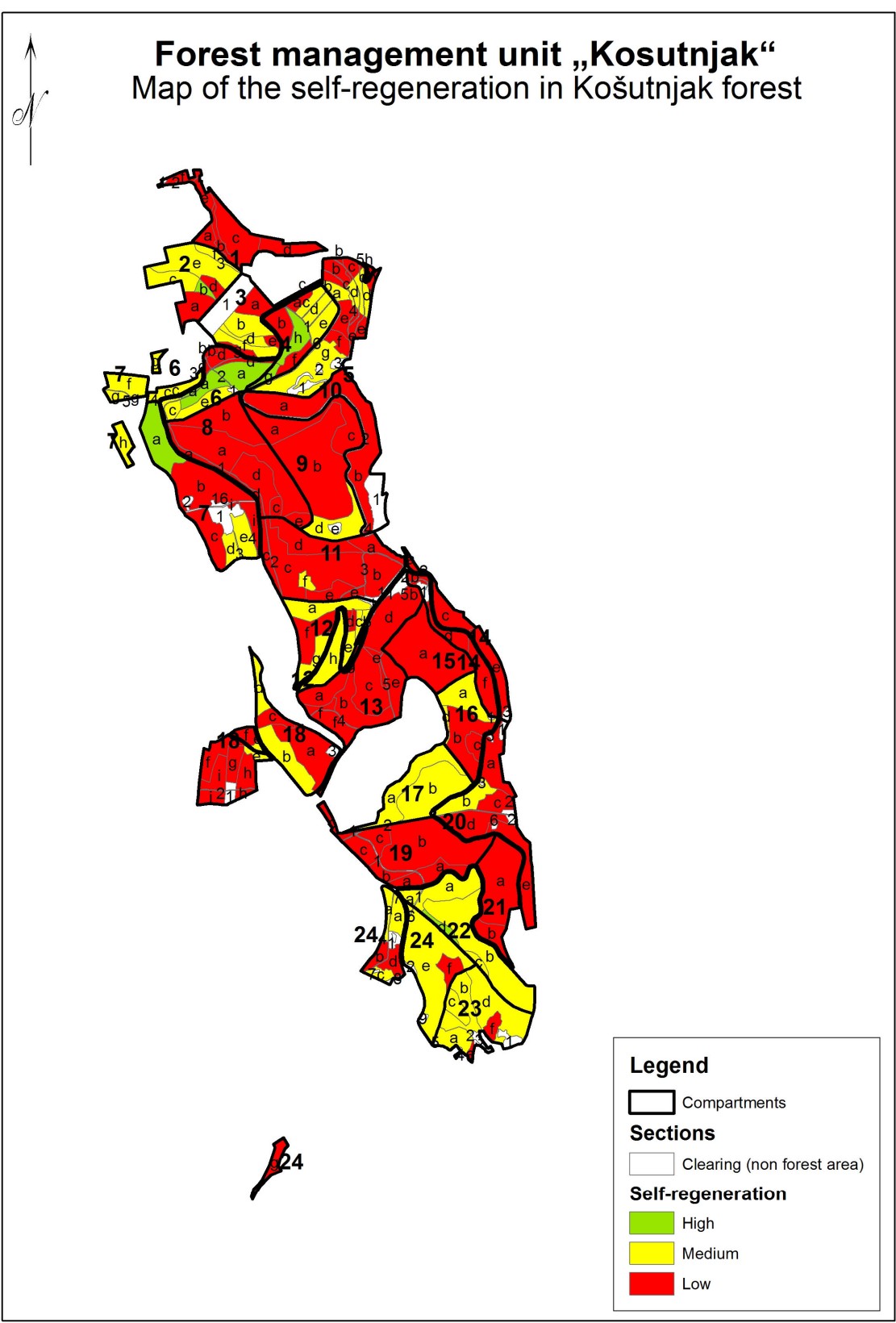

**Figure 11.** Map of self-regeneration capacity in the Košutnjak forest. Numbers—Forest department, Letters—Department subdivision.

**Table 5.** Assessment of changes, stability, and self-renewal of forests in the "Košutnjak".

| Changes | Section Number | | Stability | Section Number | | Self-Renewal | Section Number | |
|---|---|---|---|---|---|---|---|---|
| | **%** | | | **%** | | | **%** | |
| | **(ha)** | | | **(ha)** | | | **(ha)** | |
| High | 76 63.09 (163.42) | | Low | 41 21.57 (55.88) | | Low | 80 64.55 167.20 | |
| Medium | 56 36.91 (95.62) | | Medium | 88 76.72 (198.74) | | Medium | 46 31.89 (82.60) | |
| Low | 0 0 (0) | | High | 3 1.71 4.42 | | High | 6 3.57 9.24 | |
| Total | 132 100 259.04 | | Total | 132 100 259.04 | | Total | 132 100 259.04 | |

## 5. Discussion

Urban forests provide important habitats for birds, particularly for migratory species [36]. The study found that urban forests can serve as critical stopover sites for migratory birds, providing important resources for refueling and rest during their long journeys. Additionally, urban forests support a diverse range of insect populations, which are an essential food source for many bird species [37]. The study suggests that urban forests can help support healthy bird populations in urban areas by providing a reliable food source. City parks provide important ecosystem services, such as regulating temperature and air quality, which can improve the health and wellbeing of both people and wildlife in urban areas [38]. The study suggests that urban forests can help support healthy and resilient ecosystems in urban areas, which are critical for supporting biodiversity. For example, in outdoor classrooms, urban forests can provide important educational opportunities for people to learn about the value of protecting wildlife and their habitats [39]. Programs such as guided tours and educational events in urban forests can help increase public awareness about the importance of preserving these habitats for future generations.

To understand how urban forests function and how they can be effectively managed, researchers use a variety of methods to study these ecosystems. The remote sensing techniques to map and monitor urban forests over time as a method has been used in numerous studies, including a study by Imhoff et al. [40] that used satellite imagery to map and analyze urban forests in the United States. The study found that urban forests are an important component of urban ecosystems and provide important benefits such as air quality improvement and carbon sequestration. Another successful method for urban forest research is the use of citizen science programs to collect data on urban forest health and biodiversity. Citizen science programs engage members of the public in scientific research and can help increase public awareness and support for urban forest conservation. For example, a study by Tratalos et al. [41] used a citizen science program to collect data on bird species richness in urban areas. The study found that citizen science programs can be an effective way to collect data on wildlife in urban areas and can help identify areas that are important for biodiversity conservation. Locke et al. [42] used field-based surveys to measure the ecosystem services provided by urban forests in New York City. The study found that urban forests in the city provide important benefits such as carbon sequestration, air quality improvement, and stormwater runoff reduction. Overall, these methods have proven to be successful for urban forest research, providing valuable insights into the functioning and importance of these ecosystems in urban areas. By continuing to use these methods and develop new ones, researchers can help inform management decisions

and policies that support the conservation and restoration of urban forests. The work of Piras et al. [27] aims to increase knowledge about small forests (iguanas), assess the role of the landscape, and monitor their changes over the last 20 years through temporal-spatial analyses. The results show that iguanas and other small forests continue to provide an ecological network, habitat for a variety of plants and animals, firewood, and bioproducts. Maintaining traditional management of small forests is critical not only for maintaining their aesthetic purpose, but more importantly for maintaining the ecosystems associated with them. In the study by Cui et al. [28], a remote sensing ecological index (RSEI) was developed to evaluate the trend in ecological environmental changes and their driving factors in Huaibei City from 2000 to 2020. The results show that urban forest diversity indicators and physical access to nature are very important. In their study, Barrron et al. [29] use a Delphi approach to develop a set of key indicators for healthy, resilient urban forests. Two groups of experts participated in the Delphi survey: international scientists and local practitioners. As in the previously analyzed paper, the Delphi results show that urban tree diversity and physical access to nature are the most important indicators, and that "energy savings" and "forest risk" are ranked as indicators of relatively low importance. Social indicators of human health and well-being were rated particularly highly by participants.

For the applied method, the experience of forest experts with similar problems was used. Medarevic et al. [19] used a historical and analytical method. The method is essentially deductive, i.e., the existing in-formal and written sources are brought together and then analyzed, taking into account the chronology of the development of theory and practice. A similar principle was applied to our work. The experience of forestry experts with similar problems was used to shape the method applied. The same problem as ours is the reduction in forest area and the change in vegetation. Their results showed that forest management planning requires two basic assumptions: sustainability and multifunctionality. Tomićević-Dubljević et al. [26] used a survey for visitors. The aim was to raise people's awareness of the social benefits that this urban forest provides to residents. Their results showed that the majority of respondents were satisfied with the management and maintenance of the area. However, they made suggestions for improvements. Most visitors would be willing to be personally involved in the decision-making process regarding the area and showed willingness to pay a fee for maintenance. This study demonstrates the potential and need for public involvement in the management of such urban forests. The results suggest that this approach could help decision makers incorporate shared values into management decisions.

## 6. Conclusions

The research shows that through this valorization we have gained a deeper understanding of information; despite the fact that the forest has been largely altered, the stability of the forest has been maintained, so that the self-renewal of the forest is also possible to a small extent, but also, the sustainability of the forest is somewhat likely. Stand condition, stand age, composition, productivity, weed cover, health status, susceptibility, and adverse effects proved to be very important in assessing the self-regeneration capacity. Continuity of urban forest management also proved important.

It is considered that the self-renewal of the forest can be evaluated in three stages: through the analysis of the changes in the forest since the beginning of its conservation and maintenance until now, and through the analysis of the stability of the forest vegetation and soil. In assessing the likely self-renewal of the forest, a new method of forest assessment was used for this study, which was developed with the aim of demonstrating the ability of the forest to self-renew without human assistance. It is assumed that forest self-renewal can be evaluated in three stages by analyzing the changes in the forest from the beginning of its conservation and maintenance to the present day, as well as the stability of forest vegetation and soil. In assessing the likely self-renewal of the forest, the forest change assessments conducted in this study used data from a 10-year forest inventory (2007–2016). They included changes in stand composition, current stand stability, and self-renewal

capacity with a suggested forest management approach. The self-regeneration capacity of stands was estimated based on changes in the decade of study. The types of forest stands studied were determined at the level of a forest section, which serves as the basic assessment unit. The applied method helped to reach a more accurate conclusion, as the collected and evaluated data allowed a relatively simple and correct assessment of the forest condition and future development.

Some perspectives for future research arising from our research results could provide a basis for forest management, maintenance, and improvement. This work could be a useful example for other urban forests. From this work, we learned that regular monitoring, regular maintenance, and conscientious attitudes of forest users are important for forest quality.

The authors have created a framework for assessing sustainability, evaluating and mapping conditions and changes, and applied it to assessing the current state of the forest. This study is important because it can serve as a model for other urban forests in our country or it can be used as a model for repairing the condition and sustainability of all urban forests in our country.

**Author Contributions:** M.C.—corresponding author, main document arrangement, author of the EEFS methodology, preparation of tables, maps, and creation of photos; M.J.—geographical settings, creation of maps, and GIS analysis, text correction; J.T.-D.—data collection for Introduction; M.M.—co-author of EEFS methodology; L.R.—data management; V.M.—text corrections and manuscript preparation. All authors have read and agreed to the published version of the manuscript.

**Funding:** This paper presents the result of the research projects no. 451-03-47/2023-01/200027 and no. 451-03-47/2023-01/200091, funded by the Ministry of Science, Technological Development and Innovation of Republic of Serbia.

**Data Availability Statement:** Not applicable.

**Conflicts of Interest:** The authors declare no conflict of interest.

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
