# Peer review of "Ecological Evaluation of the Sustainability of City Forests"

_forests, doi:10.3390/f14040700_

Round 1

Reviewer 1 Report

Scientific research on urban forests is urgently needed, as is an evaluation of their sustainability in an anthropogenic setting. Overall, the paper is technically sound; however, some questions and comments need to be answered.

The authors claimed to have evaluated the stand capacity for self-regeneration. However, only the state of the trees was evaluated (e.g., tables 2 and 3). Why, in fact, was the generating potential of the park's trees not really assessed? The level of seed production? Renewal rate (number of undergrowth per hectare)?

Who created the age grade scale that was used to the work? When the typical lifespan of a pedunculate oak or sessile oak, for instance, is 300–400 years, how can there be mature trees that are only in the 20–40-year range?

The names of Tables 2 and 3 are identical. Verbatim! They also have nearly the same material!

As a recommendation, we suggest that the authors create a scale of recreational disturbances in urban woods based on their methodology.

Author Response

Dear Editors and Reviewers,

I am writing to submit the revised manuscript (ID forests-2255071) entitled, “ECOLOGICAL EVALUATION OF THE SUSTAINABILITY OF CITY FORESTS” for publication in Forests. I would like to thank referees for the careful and constructive reviews. Based on the comments from the referees, we have made changes of the manuscript, which are detailed below.

Reply to the evaluation by the first referee

“The authors claimed to have evaluated the stand capacity for self-regeneration. However, only the state of the trees was evaluated (e.g., tables 2 and 3). Why, in fact, was the generating potential of the park's trees not really assessed? The level of seed production? Renewal rate (number of undergrowth per hectare)?”

Response:  

Answer to the question why only the condition of the trees was evaluated.

Regarding the regional methodology of data collection in the forest stand inventory, which in urban conditions was previously based on terrestrial surveys, one of the important data for the formation of economic classes related to the determination of ceno-ecological affiliation or, in preserved stands, to the determination of forest type (regionally developed forest classification based on habitat defined by tree species and soil type). Considering the urban character of the assessed forests, they have been used spontaneously from time to time in the past (due to political crises and wars) and often significantly modified in terms of natural potential, and the capacity of the stand was sometimes estimated by analogy in relation to the stands preserved in other forest complexes. In addition to forest type as potential, recent vegetation and the relationship between these two categories were also identified as a basis for evaluating conservation and natural regeneration. Based on the altitude (up to 150 m), these forests belong to the deciduous forests in the Danube and Sava regions, to the hygrophilous poplar and willow forests, to the Pedunculate oak (Quercus robur) communities with hornbeam, linden, and, at altitudes above 100 m, to the holm oak forests. In xerothermic habitats, these are Orno-Polyqurcetum habitats. Some of these stands are called mature (> 40 years), but in terms of their functional affiliation, recreational forests.

Comment on the level of seed production:

Seed production in the habitats described above cannot rely on surviving old building trees of very poor quality that rarely bear fruit, so the plan to restore these forests is based on planting plantations in small areas with seedlings of native tree species. The hygrophilous belt of autochthonous willows and poplars (line forests) will be maintained until the physiological maturity of the dieback, as most of it is temporarily covered by NATURA 2000.

Note on renewal rate (undergrowth percentage per hectare).

It has not been determined for the reasons mentioned above, and there is also a lack of vegetative regeneration of the large-leaved linden, giving the impression of perenniality.

“Who created the age grade scale that was used to the work? When the typical lifespan of a pedunculate oak or sessile oak, for instance, is 300–400 years, how can there be mature trees that are only in the 20–40-year range?”

Response: The age scale was established by the author starting from functional maturity, which has wide limits in recreational forests and starts already at age 40 years, in willow clonal plantations at 20 years and in poplars at 25 years.

In production and commercial forests, the life span of Pedunculate oak is about 200 years and that of Sessile oak is 120-140 years. Very rare Pedunculate oak trees in Serbia are older than 150 years for many reasons.

In particular, urban forests in Belgrade survived the destruction during WW1 and WW2 and the bombing of Belgrade in 1999. With the material crisis in the 90s of the 20th century and until today, the trees still had the potential to survive on their own without the help of man. It was expected that the result of the forest change analysis would show whether the forest has changed a lot from its original composition to today in terms of its habitat potential – ceno-ecological affiliation, because it is assumed that the original forest with suitable forest vegetation is the most stable and sustainable forest.

Due to the mentioned and other reasons in the area, the average age of a mature forest stand in Košutnjak is about 80 years, if it is of natural origin, and in 1954 reforestation began along the edges of the planted forests, mostly on agricultural land expropriated from the owner.

“The names of Tables 2 and 3 are identical. Verbatim! They also have nearly the same material!”

Response: Table number 3 was a mistake. Now the correct table has been entered with the indicators of change in the forest under study. The title of the table has also been changed. (The changes to the table are visible in the text).

“As a recommendation, we suggest that the authors create a scale of recreational disturbances in urban woods based on their methodology.”

Response: This is very difficult today, as the forest is stressed by urbanization, infrastructure, aging of the remaining trees of poor quality, pronounced influence of biotic factors due to weakening immunity, desiccation, etc. We thank the experts for their suggestion, but we do not consider it necessary.

Reviewer 2 Report

I believe that some points can be improved:

- In the discussion, many initial elements could be in the description of the results, mainly those from line 347 to 388. It confused the reading. This opens up more space for your discussion.

- The discussion is weak, it could focus more on the role of urban forests as havens for biodiversity. In temperate and tropical forests there is much published material on the subject. It is worth adding more to the discussion.

- The conclusions are more for a description of justifications and presentation of the proposal than for an effective conclusion. I suggest reducing it. I believe the conclusion is right on lines 462 to 471.

Author Response

Dear Editors and Reviewers,

I am writing to submit the revised manuscript (ID forests-2255071) entitled, “ECOLOGICAL EVALUATION OF THE SUSTAINABILITY OF CITY FORESTS” for publication in Forests. I would like to thank referees for the careful and constructive reviews. Based on the comments from the referees, we have made changes of the manuscript, which are detailed below.

Reply to the evaluation by the second referee

 “- In the discussion, many initial elements could be in the description of the results, mainly those from line 347 to 388. It confused the reading. This opens up more space for your discussion.”

Response: Part of the discussion between lines 347 and 388 has been moved to the results, as you suggested, since it is certainly better placed there. This made more room for other texts in the discussion we added.

“- The discussion is weak, it could focus more on the role of urban forests as havens for biodiversity. In temperate and tropical forests there is much published material on the subject. It is worth adding more to the discussion.”

Response: At your suggestion, we have continued the discussion in the direction of urban forests as refuges for biodiversity (see text).

“- The conclusions are more for a description of justifications and presentation of the proposal than for an effective conclusion. I suggest reducing it. I believe the conclusion is right on lines 462 to 471.”

Response: At your suggestion, we have streamlined the conclusion and removed the descriptive paragraph that already appears in the methodology and results (all changes are marked in the text).

Round 2

Reviewer 1 Report

All the issues I cared about were addressed by the authors. So , in my view, the present manuscript can be accepted at the present state.

Reviewer 2 Report

Apparently all my suggestions were heeded. I have nothing else to add.